# Overexpression of BDNF and uPA Combined with the Suppression of Von Hippel–Lindau Tumor Suppressor Enhances the Neuroprotective Activity of the Secretome of Human Mesenchymal Stromal Cells in the Model of Intracerebral Hemorrhage

**DOI:** 10.3390/ijms26146697

**Published:** 2025-07-12

**Authors:** Stalik S. Dzhauari, Alexandra L. Primak, Nataliya A. Basalova, Natalia I. Kalinina, Anna O. Monakova, Kirill D. Bozov, Arkadiy Ya. Velichko, Maria E. Illarionova, Olga A. Grigorieva, Zhanna A. Akopyan, Vladimir S. Popov, Pavel G. Malkov, Anastasia Yu. Efimenko, Vsevolod A. Tkachuk, Maxim N. Karagyaur

**Affiliations:** Medical Research and Education Institute, Lomonosov Moscow State University, 27/1, Lomonosovsky Ave., 119191 Moscow, Russia; stalik.djauari@yandex.ru (S.S.D.); primak.msu@mail.ru (A.L.P.); natalia_ba@mail.ru (N.A.B.); n_i_kalinina@mail.ru (N.I.K.); monakova-anya@mail.ru (A.O.M.); kir-bozov@yandex.ru (K.D.B.); velichko.arkady@gmail.com (A.Y.V.); mar729i63illar90@yandex.ru (M.E.I.); go.grigorievaolga@gmail.com (O.A.G.); akopyanza@my.msu.ru (Z.A.A.); galiantus@gmail.com (V.S.P.); malkovp@gmail.com (P.G.M.); efimenkoay@my.msu.ru (A.Y.E.); tkachuk@fbm.msu.ru (V.A.T.)

**Keywords:** mesenchymal stromal stem cells, secretome, genetically modified cell cultures, human telomerase reverse transcriptase (hTERT), brain-derived neurotrophic factor (BDNF), urokinase-type plasminogen activator (uPA), Von Hippel–Lindau tumor suppressor (VHL), intracerebral hemorrhage (ICH)

## Abstract

Nerve tissue damage is an unsolved problem in modern neurology and neurosurgery, which prompts the need to search for approaches to stimulate neuroprotection and regeneration of neural tissue. Earlier we have shown that the secretome of human mesenchymal stromal cells (MSCs) stimulates rat survival, reduces the severity of neurological deficits, and decreases the volume of brain damage in a hemorrhagic stroke model. A significant disadvantage of using the MSC secretome is the need to concentrate it (at least 5–10 fold) to achieve appreciable pharmacological activity. This increases the cost of obtaining clinically applicable amounts of secretome and slows down the clinical translation of this technology. Here, we created a number of genetically modified human MSC cultures, including immortalized MSCs and those with hyperexpression of brain-derived neurotrophic factor (BDNF) and urokinase-type plasminogen activator (uPA) and with suppressed expression of Von Hippel–Lindau tumor suppressor (VHL), and we evaluated the pharmacological activity of their secretomes in a model of intracerebral hemorrhage (ICH) in rats. The secretome of MSCs immortalized by hyperexpression of the catalytic subunit of human telomerase (hTERT) revealed neuroprotective activity indistinguishable from that of primary MSC cultures, yet it still required 10-fold concentration to achieve neuroprotective efficacy. The secretome of MSC culture with combined hyperexpression of BDNF and uPA and suppressed expression of Von Hippel–Lindau tumor suppressor even without additional concentration reduced the severity of neurological disorders and decreased brain lesion volume in the ICH model. The secretomes of MSCs with separate overexpression of BDNF and uPA or suppression of VHL had no such effect or, on the contrary, revealed a toxic effect in the ICH model. Presumably, this may be due to an imbalance in the representation of individual growth factors in the secretome of genetically modified MSCs, which individually may lead to undesirable effects in damaged nervous tissue, such as increased permeability of the blood–brain barrier (under the influence of pro-angiogenic factors) or neural cell apoptosis (due to an excess of neurotrophic factors). The obtained data show that genetic modification of MSC cultures can enhance or alter the therapeutic activity of their secretomes, which can be used in the creation of promising sources of biopharmaceutical substances.

## 1. Introduction

The search for approaches to stimulate neuroprotection and regeneration of damaged neural tissue is one of the main and still unsolved problems in neuropharmacology and regenerative biomedicine [1,2,3]. We have previously shown that the secretome of human mesenchymal stromal cells (MSCs), which contains an abundance of growth factors, cytokines, matrix proteins, and extracellular vesicles, stimulates rat survival, reduces the severity of neurological deficits, and decreases the brain lesion volume in a model of intracerebral hemorrhage (ICH) in rats [4].

The secretome of MSCs seems to be very promising for the stimulation of regenerative processes due to its complex biomimetic activity. Its significant disadvantages are the short lifespan of the producer cells (MSCs) in culture and heterogeneity of their properties from donor to donor [5,6,7,8], as well as the necessity to concentrate the obtained secretome at least 5–10 fold to achieve appreciable pharmacological activity (in most in vitro and in vivo models) [4,9]. A possible way to overcome these limitations is a genetic modification of MSCs in order to prolong the lifetime of the producer culture, stabilize the composition of their secretome, or increase the content of key active components (growth factors, microRNAs, etc.) in the secretome.

Hyperexpression of human telomerase reverse transcriptase (hTERT) in MSC cultures delays their aging and prolongs the time of their active use up to 38–45 doublings of cell population (25–30 passages), which was demonstrated earlier [10,11,12]. At the same time, immortalization of MSC cultures practically does not change the composition and biological activity of their secretomes, stabilizes the content of the main pro-angiogenic and neurotrophic factors, and prevents the appearance of senescence-associated secretory phenotype (SASP) components in the secretome up to 30 cell doublings (20 passage) [10]. Whether the secretomes of immortalized MSC cultures (iMSCs) are an equivalent substitute for the secretome of primary isolated MSCs (pMSCs) for stimulation of neuroprotection after ICH remains unknown. Verification of the pharmacological activity of the iMSC secretome was one of the objectives of this study.

The need to concentrate the MSC secretome 5–10 fold to increase its biological activity is a limitation for this technology and it can be overcome through the hyperexpression of key secretome components that mediate its pro-regenerative activity. Here, we used two different approaches to accomplish this task: (1) expression of recombinant growth factors (brain-derived neurotrophic factor, BDNF, and urokinase-type plasminogen activator, uPA) through direct delivery of their genes into MSCs, and (2) enhancement of endogenous growth factor production (primarily pro-angiogenic growth factors) through the suppression of VHL and stabilization of HIF-1a. BDNF and uPA were chosen as molecules for overexpression in MSC culture because BDNF is one of the main neurotrophic factors stimulating survival and regeneration in a wide range of neurons [13,14] and uPA has pleiotropic effects, including activation of growth factors, stimulation of angiogenesis and neuritogenesis, triggering of the blood anti-clotting system, etc. [15,16]. Moreover, we have previously demonstrated that BDNF and uPA functionally complement each other, stimulating neuroprotection in the ICH model [17].

An alternative to direct modification of MSC cultures with growth factor genes is the strategy of increasing the production of endogenous factors through the activation of signaling cascades proposed by nature itself, for example, through simulating hypoxia of a cell culture by suppressing VHL and stabilizing and increasing the activity of HIF-1a [18,19].

VHL (Von Hippel–Lindau) is a negative master regulator of HIF proteins, a family of transcription factors [20,21], inducing cell metabolism adaptation and expression of a wide range of pro-angiogenic molecules (VEGF, erythropoietin, angiopoietins 1/2, PlGF, HGF, etc.) in response to hypoxia [22,23]. According to literature data, suppression of VHL in a number of tumor cells leads to constitutive activation of HIF molecules (regardless of hypoxia), which in turn increases the expression of the mentioned pro-angiogenic factors [24,25,26], a number of which have distinct neuroprotective and pro-regenerative activities [27,28]. We supposed that induction of the expression of a wide range of growth factors that complement each other and are expressed in physiologic ratios may be an even more advantageous strategy in biopharmacy than direct forced gene delivery of individual, even extremely important, growth factors into producer cells. Here, we tried to evaluate the efficiency of the selected approaches to improve the pharmacological properties of the MSC secretome and enhance its neuroprotective activity in an ICH model in rats.

## 2. Results

### 2.1. Genetic Modification of iMSCs Increases the Content of Growth Factors in Their Secretome

To increase the content of growth factors stimulating neuroprotection and neural tissue regeneration in the secretome of iMSCs, we developed a number of vectors for their constitutive or inducible expression. To increase the expression of BDNF and uPA, we used direct delivery of their genes into iMSCs, and to increase the expression of pro-angiogenic factors, we used shRNA-mediated suppression of VHL and stabilization of HIF-1a (Figure 1).

The secretome of iMSCs without additional genetic modification was characterized previously [10]. In this study, the obtained lot of iMSC secretome contained 0.73 ± 0.16 ng/mL BDNF, 1.65 ± 0.29 ng/mL uPA, 0.66 ± 0.19 ng/mL VEGF, and 2.93 ± 0.57 ng/mL HGF (*n* = 3).

Before assessing the content of growth factors in the secretome of genetically modified iMSCs, it was necessary to determine the optimal doxycycline concentrations providing the highest level of recombinant protein/RNA production in a cell culture with induced expression. We titrated doxycycline concentrations of 0, 20, 80, 200, and 500 ng/mL, followed by ELISA analysis of the culture medium for BDNF and uPA contents. It was found that the induced expression systems provided high levels of BDNF and uPA production, reaching a “plateau” in the range of doxycycline concentrations of 80–500 ng/mL. Thus, at a doxycycline concentration in the culture medium of 200 ng/mL, the BDNF content in the secretome and MSCs increased ~22 fold from 0.74 ± 0.14 ng/mL to 16.22 ± 4.07 ng/mL, and the uPA content increased ~2 fold from 2.11 ± 0.44 ng/mL to 4.26 ± 0.78 ng/mL (* *p* < 0.05; *n* = 3; one-way ANOVA) (Figure 2).

The differences in the increases in BDNF and uPA expression (~22 fold vs. ~2 fold, respectively) within this inducible system are explained by a combination of two facts. First, uPA is initially expressed at a high baseline level (3 times higher than BDNF), which masks minor changes. Second, the structural features of the expression cassette of the Len-ti-tetO-BDNF-IRES-uPA vector suggest more efficient translation of the first transcript (BDNF) within the bicistronic RNA via the 5′-cap-dependent mechanism compared to translation of the second transcript (uPA) via the IRES-dependent mechanism [30,31].

We did not observe any noticeable cell death in iMSC cultures at the doxycycline concentration of 500 ng/mL, which correlated with previously published data [32]. Therefore, we suppose that the plateau effect in the expression of growth factors with an increase in the concentration of doxycycline may be explained not by the toxic and cytolytic effect of doxycycline but by the saturation of all binding sites of the doxycycline transactivator within the Lenti-tetO-BDNF-IRES-uPA vector.

In order to ensure and maintain a supramaximal level of recombinant protein production in the inducible expression systems (taking into account the half-life of doxycycline of 24 h under the culture conditions used), we chose a doxycycline concentration of 200 ng/mL, which provided a slightly higher production of recombinant BDNF compared to the doxycycline concentration of 80 ng/mL (although no statistically significant differences in BDNF and uPA concentrations were determined between these groups) (Figure 2). The doxycycline concentration of 200 ng/mL, as supramaximal, was further used to induce secretome production in iMSC cultures with induced expression of BDNF and uPA, as well as anti-VHL-shRNAs, in order to establish the neuroprotective activity of these modified secretomes in the ICH model.

Comparative analysis of the efficiency of growth factor expression by the iMSC cultures modified with the constitutive and inducible expression cassettes showed that the highest levels of BDNF and uPA were observed in the cultures with inducible expression (Figure 3A, Table 1), which correlated with the mRNA levels of these factors in the appropriate iMSC cultures (Figure 3B). Thus, the secretome of iMSCs inducibly expressing BDNF contained significantly more BDNF than those constitutively expressing it: 19.3 ± 4.2 ng/mL vs. 4.9 ± 1.3 ng/mL (*p* < 0.005; *n* = 3; *t*-test), respectively, and significantly higher than the secretome of unmodified iMSC (*p* < 0.001; *n* = 3, one-way ANOVA). The levels of uPA in the secretomes of iMSCs with constitutive and inducible expression cassettes were 3.3 ± 0.83 ng/mL and 4.9 ± 0.98 ng/mL, respectively, both of which were significantly higher than that in the secretome of unmodified iMSC (*p* < 0.05; *n* = 3; one-way ANOVA).

The inducible shRNA expression cassette also provided a higher level of anti-VHL shRNA compared to the use of constitutive expression system and was significantly different from that in unmodified iMSCs (*p* < 0.05; *n* = 3; one-way ANOVA; Figure 3C). This resulted in 2.9- and 5.3-fold decreases in VHL mRNA levels in MSCs with constitutive and inducible shRNA expression, respectively. However, VHL suppression had very little effect on the expression levels of pro-angiogenic factors VEGF and HGF. Small and statistically insignificant increases in VEGF and HGF concentrations by 1.2–1.3 fold were observed only in the secretome of iMSCs inducibly expressing the shRNA. Thus, the VEGF concentration in the secretome of such iMSCs increased up to 0.86 ± 0.19 ng/mL, and the HGF concentration increased up to 3.6 ± 0.63 ng/mL (Figure 3A, Table 1). Constitutive expression of anti-VHL shRNA did not cause any significant increases in the VEGF and HGF concentrations in the iMSC secretome.

It should be noted that additional genetic modification of iMSC cultures with cassettes of constitutive or inducible expression of growth factors/shRNA resulted in the slowing down of the iMSC proliferation rate, and it was most prominent in MSC cultures carrying constitutive expression cassettes, which imposes certain limitations on the possibility of practical use of such cultures in the future.

The human pMSC secretome used in this study contained 0.7 ± 0.16 ng/mL BDNF, 1.3 ± 0.28 ng/mL uPA, 0.53 ± 0.06 ng/mL VEGF, and 2.62 ± 0.49 ng/mL HGF (*n* = 3), and it was not significantly different from the secretome of unmodified iMSCs in terms of the content of these factors.

An interesting result of this study is the fact that doxycycline induction statistically significantly increased the contents of BDNF and uPA mRNA, as well anti-VHL shRNA, in the secretome of MSCs bearing the inducible Lenti-tetO-BDNF-IRES-uPA and Lenti-tetO-shRNA expression cassettes (Figure 3D). However, the compartmentalization (soluble secretome fraction or extracellular vesicles) and physiological activity of the identified shRNAs and mRNAs of BDNF and uPA are yet to be established.

### 2.2. iMSC and pMSC Secretomes Reveal Equal Neuroprotective Activity in the ICH Model

The secretome of unmodified iMSCs concentrated 10 fold (iMSC10x) stimulates neuroprotection of brain tissue in the ICH model, similarly to what was shown previously for the secretome of pMSCs [4,9]. Thus, the iMSC secretome increases survival of experimental animals, reduces the severity of neurological disorders in them, and decreases the brain lesion volume, according to MRI. In the iMSC group, the survival rate of experimental animals during all 14 days of the experiment reached 100% and, 10 days after modeling of ICH, all rats looked healthy with no signs of neurological disorders. At the same time, in the control group, where rats were injected with empty culture medium (DMEM-LG) concentrated 10 fold, 10 days after ICH modeling, the survival rate was 83% (5 of 6), and 17% of rats (1 of 6) had mild neurological disorders. The mean volume of the brain lesion focus in rats injected with iMSC10x secretome was 134 (66; 161) mm^3^, whereas the animals injected with pMSC secretome or DMEM-LG had mean brain lesion volumes of 121 (78; 147) and 188 (159; 256) mm^3^, respectively (Figure 4) (*p* < 0.05; n ≥ 6; ANOVA on ranks: Dunn’s criterion).

### 2.3. Overexpression of BDNF and uPA Combined with the Suppression of VHL Enhances the Neuroprotective Activity of iMSC Secretome in the Model of Intracerebral Hemorrhage

Hyperexpression of hTERT prolongs the lifetime of iMSC cultures and stabilizes the composition of their secretome but keeps the necessity to concentrate the iMSC secretome (as well as the pMSC secretome) at least 10 fold to achieve a noticeable neuroprotective effect due to relatively low content of molecules with neuroprotective activity.

To increase the neuroprotective activity of the secretome, we developed a number of genetically modified MSC lines based on the previously obtained iMSC culture [10]. The neuroprotective activity of the secretomes of the obtained iMSC cultures in different combinations, with different contents of neuroprotective and angiogenic molecules (without any concentration procedure), was evaluated in the previously described ICH model in rats [33]. The analysis showed that the most noticeable neuroprotective effect was observed in rats treated with a combination of secretomes from iMSCs with constitutive expression of BDNF + uPA (cBU) and anti-VHL shRNA (cSh) in a 1:1 ratio, containing 3.25 ± 0.45 ng/mL BDNF, 2.28 ± 0.40 ng/mL uPA, 0.59 ± 0.15 ng/mL VEGF, and 2.61 ± 0.31 ng/mL HGF (Table 1). The cBU + cSh combination increased rat survival up to 100% (5 of 5) and reduced the brain lesion volume to 104 (86; 125) mm^3^, whereas in the control group (DMEM-LG), the survival rate of experimental animals was 80% (4 of 5) and the mean brain lesion volume was 187 (151; 226) mm^3^ (Figure 5). No significant differences in the volume of brain lesion foci between studied groups were observed due to the small sample size and multiple experimental groups; however, there was a trend toward lower lesion volume in the cBU + cSh group. Unconcentrated (1x) iMSC secretome containing 0.73 ± 0.16 ng/mL BDNF, 1.65 ± 0.29 ng/mL uPA, 0.66 ± 0.19 ng/mL VEGF, and 2.93 ± 0.57 ng/mL HGF (Table 1) also increased rat survival (5 out of 5) but did not affect the lesion foci volume (165 (153; 213) mm^3^).

The iBU + iSh combination, similarly to the cBU + cSh combination, had a prominent neuroprotective effect, manifested in stimulation of rat survival (5 out of 5), decreased severity of neurological deficits, and decreased volume of brain damage to 108 (99.3; 174) mm^3^. At the same time, the secretome of genetically modified iMSCs expressing BDNF+ uPA or anti-VHL shRNA used separately exerted less noticeable or even toxic effect. Thus, the secretome of iMSCs with induced expression of BDNF + uPA (iBU), containing 19.32 ± 4.2 ng/mL BDNF, 4.94 ± 0.99 ng/mL uPA, 0.65 ± 0.19 ng/mL VEGF, and 2.67 ± 0.59 ng/mL HGF (Table 1), decreased the survival of rats to 60% (3 out of 5) and vice versa increased in the lesion focus volume to 232 (210; 238) mm^3^. These parameters in the iBU group were even worse than in the DMEM-LG control group.

Similarly, the secretome of iMSCs with constitutive BDNF and uPA overexpression (cBU), despite higher contents of BDNF and uPA than in the secretome of unmodified iMSCs (iMSC1x) (4.90 ± 1.31 ng/mL and 3.33 ± 0.84 ng/mL vs. 0.73 ± 0.16 ng/mL and 1.65 ± 0.29 ng/mL, respectively) did not reveal any neuroprotective activity in the ICH model. The brain lesion volume in cBU rats reached 212 (176; 253) mm^3^. The histological analysis of brain slices stained with hematoxylin/eosin confirmed the data obtained during MRI-study (Figure 5).

## 3. Discussion

Human mesenchymal stromal cells are powerful stimulators of tissue regeneration and repair processes, and the totality of their secretion products (secretome) may represent a promising platform for the development of biopharmaceutical drug candidates, including those for the stimulation of neuroprotection and regeneration of injured nervous tissue [34,35].

At the same time, primary isolated MSCs (pMSCs) are an inconvenient and extremely limited source of secretome. This is due to a number of parameters, including the lack of donors, rapid aging of MSC cultures, the heterogeneity of the qualitative and quantitative compositions of their secretome, as well as the need to control the infectious safety and biological activity of each newly obtained sample [33].

The results of an analysis of the world literature reveal that only products (e.g., secretome) of standardized characterized cell lines, such as immortalized MSCs (iMSCs), have real prospects of entering clinical practice [33,36,37]. Immortalization allows us to overcome all of the listed limitations, and the stability of the qualitative and quantitative compositions of the secretome after immortalization was confirmed earlier [10]. However, whether the secretome of immortalized MSCs retains the ability to stimulate neuroprotection of brain tissue remained unknown.

Hyperexpression of hTERT in MSC culture slows down cell aging and significantly increases the time interval (at least from 8 to 20 passages) for secretome production, which in theory allows obtaining hundreds of tons of secretome with relatively constant qualitative and quantitative compositions [10]. However, the secretomes of pMSCs and iMSCs must be concentrated to achieve a noticeable pharmaceutical effect. The iMSC and pMSC secretomes, concentrated 10 fold, stimulate the survival of experimental animals and reduce the brain lesion volume in the ICH model, even after a single administration. The obtained data allowed us to consider iMSC cultures as a promising source of secretome for biopharmaceutical production, which is especially relevant in view of the rapid expansion of the areas of potential application of cell-based and cell-free therapy [33,38,39].

The need to concentrate the secretome of pMSCs or iMSCs to increase its pharmacological activity is associated with a number of disadvantages. On the one hand, it may alter the ratio of the components. Moreover, it increases the labor and cost of pharmacological production, with a general decrease in the yield of biopharmaceutical substances.

Potentially, this limitation can be overcome by increasing the expression in iMSC cultures of factors stimulating tissue protection and regeneration [40,41]. Here, to enhance the pro-regenerative activity of the iMSC secretome, we used approaches to increase the expression of pro-angiogenic molecules, as well as factors with previously demonstrated neuroprotective activity: brain-derived neurotrophic factor (BDNF) and urokinase-type plasminogen activator (uPA) [13,14,15,16,17]. For this purpose, we created systems of constitutive and inducible expression of the combination of BDNF and uPA genes and suppression of the VHL gene (an HIF-1a suppressor).

We previously identified BDNF as one of the main growth factors responsible for the neuroprotective properties of the MSC secretome [42]. This was confirmed by other authors [43,44,45]. At the same time, the neuroprotective activity of BDNF by itself is inferior to the combined action of the neuroprotective and pro-regenerative factors in the MSC secretome. Combining BDNF with other functionally complementary factors (including uPA) potentiates its neuroprotective activity, which was shown earlier [17], and, apparently, underlies the neuroprotective activity of BDNF in the MSC secretome.

Suppression of VHL (a negative regulator of HIF-1a) expression by constitutive or inducible expression of anti-VHL shRNA was designed to stabilize HIF-1a and increase the expression of a wide range of pro-angiogenic factors. However, despite the 2.9- and 5.3-fold decreases in VHL mRNA levels in iMSC cultures with constitutive and inducible shRNA expression, respectively, no significant increases in the levels of pro-angiogenic factors, such as VEGF, HGF, and uPA, in the secretome of these iMSCs were achieved. The non-significant increases in VEGF and HGF concentrations by 30% and 23%, respectively, were observed only in the group with induced expression of anti-VHL shRNA (compared to the secretome of unmodified iMSCs). At the same time, a detectable amount of anti-VHL shRNA was found in the secretome of iMSCs with induced expression of shRNA, which was consistent with the data of Rossella Crescitelli et al., who reported that expressed RNAs can be found in extracellular vesicles produced by cell culture [46,47,48]. Similarly, detectable amounts of BDNF and uPA mRNA were found in the secretome of iMSC cultures with inducible expression of BDNF and uPA. This phenomenon allows us to consider the MSC secretome as a preparation that combines the advantages of recombinant protein therapy (immediate activation of anti-apoptotic cascades in the focus of acute damage, which is especially important for stimulation of neuroprotection) and gene therapy (nature-like dynamics of expression and concentration gradient of therapeutic molecules), providing complex stimulation of the regeneration process [49].

Hyperexpression of recombinant proteins was not traceless for cell cultures—it significantly reduced the proliferation rate of producer cells. In particular, it was especially noticeable for iMSC cultures with constitutive expression of BDNF and uPA (cell culture doubling time ~80 h), which limits the possibility of using such modifications to create industrially applicable cell lines. The mechanism of this phenomenon has not been established. Presumably, it may be due to overloading of the cell synthetic apparatus by the synthesis of recombinant proteins or the action of the secreted BDNF and/or uPA proteins themselves on the cell culture. The doubling time of cultures with induced expression of BDNF and uPA (beyond induction) also increased (~60 h), although to a lesser extent than for constitutively expressing iMSCs.

Non-concentrated secretomes of genetically modified iMSC cultures demonstrated a wide range of neuroprotective potency in the ICH model: from neuroprotective to toxic. Thus, secretomes with elevated contents of BDNF and uPA did not reduce brain lesion volume, and a tendency for such an effect to depend on the concentration of BDNF was observed. This contradicts earlier data on the key role of BDNF in the realization of neuroprotective activity of MSCs [42,43,44,45] and can be explained by the dependence of the effect on the dose, route of administration, dynamics of BDNF expression, as well as modulation of its activity by other factors present in the secretome. BDNF is known to be secreted as a mixture of pro-BDNF and mature BDNF [13,14]. Excess BDNF or the predominance of its pro-form may activate the pro-apoptotic and pro-neuroinflammatory p75NTR-ceramide signaling pathway [13,14,50,51,52]. However, the mechanism of the toxic effect of the secretome of iMSC culture overexpressing BDNF and uPA administered intravenously in the ICH model remains to be established.

Hematoxylin/eosin staining of rat brain sections revealed an increased volume of necrotic tissue in the cBU and iBU groups, but no evidence of significant leukocyte infiltration. Establishing the mechanism of the toxic effect of high-dose BDNF/uPA requires additional studies. Possible mechanisms include increased neuroinflammation, impaired blood–brain barrier function, or direct toxic effects of excessive amounts of growth factors. Our previously published data revealed that uPA can stimulate microglial activation when administered intracerebrally [17]. Which effect (pro-regenerative or toxic) will be predominant is apparently determined by the concentration of uPA and the duration of its action.

Similarly, no significant neuroprotection was observed in animal groups where the secretomes of iMSC cultures with anti-VHL shRNA expression (in order to increase the expression of pro-angiogenic factors (in particular, VEGF and HGF) through HIF-1a stabilization) were administered. However, a toxic effect was not observed either, although it is known that VEGF can increase the permeability of the blood–brain barrier (BBB), causing brain edema and worsening the prognosis in brain and spinal cord injuries [53,54]. This may be due to the relatively low concentrations of VEGF in the tested secretomes, which did not reach toxic levels.

An unexpected neuroprotective effect was observed in animal groups administered the combinations of secretomes with overexpressed BDNF + uPA and anti-VHL shRNA (cBU + cSh and iBU + iSh), manifested by a decrease in the severity of neurological deficits and a tendency to diminish the brain lesion volume. Although the differences observed were not statistically significant due to the small sample size, similar neuroprotection enhancement was present in both groups with secretome combinations (cBU + cSh and iBU + iSh).

The mechanism of this phenomenon has not been established and requires further study. A comparison of the results of the in vivo study and the content of growth factors in the obtained secretomes allows us to state that the observed phenomenon was not a consequence of direct dilution of high and toxic concentrations of BDNF and/or uPA. We assume that this phenomenon serves as an example of cooperative activity of neurotrophic and pro-angiogenic factors complementing each other. Presumably, pro-angiogenic factors, particularly VEGF, may increase the permeability of the BBB and enhance the delivery of neuroprotective molecules like BDNF and uPA to the site of injury or shift the balance of activation between the pro-apoptotic p75NTR-ceramide and the neuroprotective TrkB-PI3K/Akt signaling pathways in favor of the latter [55,56,57]. The investigation of specific mechanisms of possible cooperative activity of neurotrophic and angiogenic factors requires additional studies involving inhibitory, transcriptomic, or signalomic analysis.

We also cannot exclude that this phenomenon may be due to the action of coding and/or regulatory RNAs present in the secretome. The therapeutic activity of secretome mixtures (iBU + iSh and cBU + cSh) requires further studies of their effectiveness in dose-dependence and isobologram analysis, which may be the prospectives of this research.

The data obtained reaffirm that the MSC secretome contains molecules with both synergistic and antagonistic activities. The resulting therapeutic effect of the secretome is largely determined by the ratio and interactions of these molecules with each other and with the recipient’s cells and molecules. From this point of view, the MSC secretome is a convenient object for studying the interactions of growth factors in the processes of tissue renewal and regeneration. The use of the MSC secretome as a platform for developing drugs in regenerative biomedicine requires stabilization of its qualitative and quantitative composition, which can be achieved through the immortalization or genetic modification of producer cells. Our data indicate that the secretomes of immortalized MSC cultures represent a promising and commercially viable alternative to that of primary isolated MSCs. The therapeutic properties of the secretome of genetically modified MSC cultures can vary significantly—from promoting regeneration to inducing toxic effects—depending on the specific modifications made to the producer cells. This variability necessitates careful and thorough evaluation of the secretome’s properties in each case.

## 4. Materials and Methods

### 4.1. Cell Cultures

Primary isolated human mesenchymal stromal cells (pMSCs) were obtained from the biobank of the Institute of Regenerative Medicine, Lomonosov Moscow State University (https://human.depo.msu.ru (accessed on 10 July 2025)) and cultured in Advance Stem Cell Basal Medium (HyClone, South Logan, UT, USA, #SH30879.02) containing 10% Advance Stem Cell Growth Supplement (HyClone, USA, #SH30878.01) and 1x antibiotic/antimycotic mixture (Gibco, Grand Island, NY, USA, #15240062). All procedures with patient tissue samples were performed in accordance with the Declaration of Helsinki and approved by the Ethical Committee of Lomonosov Moscow State University (#IRB00010587), protocol #4 (2018). Immortalized MSC cultures (iMSCs) were previously obtained and characterized by our team [10].

MSC cultures were characterized as plastic-adhering cells expressing CD73, CD90, and CD105, deprived of the expression of hematopoietic and endothelial markers (CD14, CD19, CD20, CD34, CD45, and HLA-DR), and capable of differentiating in vitro into adipogenic, osteogenic, and chondrogenic lineages, which aligns with the criteria established by the International Society for Cell Therapy (ISCT) [58,59].

HEK293T cells (ATCC, #CRL-3216™) were cultured in DMEM high glucose medium (Gibco, Grand Island, NY, USA, #11965092) supplemented with 10% fetal bovine serum (FBS) (Gibco, Grand Island, NY, USA, #26140079) and 1x antibiotic/antimycotic mixture (Gibco, Grand Island, NY, USA, #15240062). Lentiviral particles were assembled in the same medium but with 2% FBS.

All cell cultures were cultured at 37 °C under 5% CO_2_ in a Binder CB150 incubator (BINDER GmbH, Tuttlingen, Germany). The culture medium was changed every 3–4 days.

### 4.2. Animals

The neuroprotective activity of the pMSC and iMSC secretomes in the model of intracerebral hemorrhage was assessed in male Wistar rats aged 3–3.5 months (weight 280–340 g). A single animal was an experimental unit in this study.

The animals were obtained from the Scientific and Production Enterprise “Laboratory Animal Nursery” Branch of the Institute of Bioorganic Chemistry of the Russian Academy of Sciences (Pushchino, Russia, https://www.spf-animals.ru/labanimals/rats/ (accessed on 10 July 2025)). The animals were housed and the experiment was performed in the SPF vivarium of the Medical Research and Education Institute of Lomonosov Moscow State University. The animals had normal immune status, were healthy, had no genetic modifications, and had not been manipulated before the study. The rats were examined by a veterinarian before the start of the experiment.

Animals were kept and used in the experiment in full compliance with Directive 2010/63/EU (20 October 2010) and the recommendations of the Lomonosov Moscow State University Bioethics Committee (approvals #3.4 dated 21 March 2021 and #3.5 dated 17 March 2022).

### 4.3. Assembly of Vectors for MSC Genetic Modification

The transfer vectors for lentiviral particle (LVP) assembly were assembled using standard molecular biological techniques based on the Lenti vector (Takara, USA): Lenti-Ef1a-shRNA, Lenti-tetO-shRNA, Lenti-Ef1a-BDNF-IRES-uPA, and Lenti-tetO-BDNF-IRES-uPA. A pair of short hairpin RNAs (shRNA) was used to suppress VHL gene activity, 5′-AAACAGUAGUCCAGGCUACUACUCCAUCACAGCAUGGAGUAGUAGCCUGGACUGGACUGUUUUUUUCC and 5′-AAUCCAUUUGGUAGGUAGGACCAGCACAGCACAGCCUCUGGUCCUCCUACCAAUGGAUUUUCC, expressed under the control of constitutive (Ef1a) and inducible (tetO) promoters. The shRNA expression cassette was assembled from synthetic oligonucleotides (Appendix A), and the BDNF-IRES-uPA expression cassette was described by our team previously [31]. The genetic construct FUdeltaGW-rtTA encoding a doxycycline transactivator was a generous gift from Dr. Konrad Hochedlinger [33].

### 4.4. Genetic Modification of MSC

The LVPs were assembled as described previously with some modifications [60]. Transduction of iMSC cultures was performed at 7–8 passages. For this purpose, iMSCs were seeded on 6-well plates with the confluence reaching 30–40%. For transduction, a portion of medium (1.2–1.5 mL) containing the appropriate type of LVP (MOI ~ 20) mixed with protamine sulfate (50 μg/mL) was added to each well. The plate was then incubated in the CO_2_ incubator for 1.5–2 h, after which the medium was changed with a fresh portion of LVP/protamine sulfate-containing medium. The transduction procedure was repeated 4 times for each MSC culture.

Vectors for the hyperexpression of growth factors or shRNA to VHL mRNA did not contain additional selection cassettes in order to limit the vector size (<6 kbp) to increase the efficiency of LVP assembly and transduction [61,62]. Transduction efficiency was controlled by co-transduction with an LVP encoding the BFP (blue fluorescent protein) gene in a LVP_BU/shRNA_ to LVP_BFP_ ratio = 4:1. A week after the 4 rounds of transduction, fluorescence in the BFP channel in all genetically modified iMSC cultures exceeded 80%.

### 4.5. Obtaining MSC Secretomes

The MSC secretomes were obtained as described earlier [63]. Briefly, subconfluent cultures of human pMSCs (5–7 passages), iMSCs (13–15 passages), or genetically modified iMSCs (12–14 passages) were thoroughly washed with Hanks’ solution (PanEco, Moscow, Russia, #P020P) and then cultured for 7 days in DMEM low glucose medium (Gibco, Grand Island, NY, USA, #31885049) supplemented with GlutaMAX™, pyruvate, and 100 U/mL penicillin-streptomycin (Gibco, Grand Island, NY, USA, #15240062).

The optimal concentration of doxycycline for inducing the expression of recombinant molecules was determined by titrating its concentration (0, 20, 80, 200, and 500 ng/mL) in the culture medium of iMSCs transduced with Lenti-tetO-BDNF-IRES-uPA, measuring the production of BDNF and uPA. Doxycycline was added to the cell culture medium at the appropriate concentration every 48 h to compensate for its degradation. At the end of the conditioning period, the medium was collected and centrifuged for 10 min at 300× *g* to remove cellular debris, and then concentrated 10 fold, if needed, using ultrafiltration cartridges with a 10 kDa molecular weight cutoff (Merck, Darmstadt, Germany). Protein loss during filtration was not precisely assessed. The procedure of concentrating the pMSC and iMSC secretomes was carried out in parallel using concentration cartridges from the same manufacturer.

### 4.6. Enzyme-Linked Immunosorbent Assay

The concentrations of BDNF, uPA, VEGF, and HGF in MSC secretomes were assessed using the Human Free BDNF Quantikine ELISA Kit (R and D, #DBD00), Human uPA ELISA Kit (URK) (Abcam, Waltham, MA, USA, #ab119611), Human VEGF Quantikine ELISA Kit (R&D, #DVE00), and Human HGF Quantikine ELISA Kit (R&D, #DHG00), respectively, according to the manufacturers’ instructions.

### 4.7. Reverse Transcription and qPCR

Total RNA was isolated using the Direct-zol RNA Miniprep kit (Zymo Research, Irvine, California, USA, #R2052), according to the manufacturer’s instructions. For shRNA analysis, total RNA was treated with *E. coli* Poly(A) Polymerase (#M0276S, NEB, Ipswich, MA, USA), according to the manufacturer’s instructions, and then used for cDNA synthesis. All RNA handling was performed on ice, using RNAse-free purity-grade consumables. cDNA synthesis was performed using the MMLV RT kit (Eurogen, Moscow, Russia, #SK021), according to the manufacturer’s recommendations. Oligo (dT)15 primer (Eurogen, Moscow, Russia, #SB001) was used as a primer for reverse transcription. Aliquots of 1–2 µg of RNA (depending on the RNA concentration) per 20 µL of reaction mixture were used as a matrix. The resulting cDNA was used immediately after preparation or stored at −80 °C for no more than 1 week. Real-time PCR was performed using 5X qPCRmix-HS SYBR mix (Eurogen, Moscow, Russia, #PK147L), according to the manufacturer’s instructions. A volume of 2 μL of synthesized cDNA was added to the reaction. The primers used and amplification parameters are given in Appendix A.

### 4.8. A Model of Intracerebral Hemorrhage

The neuroprotective activity of the MSC secretomes was studied using the model of intracerebral hemorrhage (hemorrhagic stroke) described by A.N. Makarenko et al. [64]. Stroke is modeled by injecting autologous blood into the internal capsule of the brain, which includes the nerve tracts connecting the cortex with the basal ganglia, spinal cord nuclei, and cranial nerve nuclei. Damage to the internal capsule causes neurological deficits, the severity of which correlates with the severity of injury.

To model ICH, rats were anesthetized with a solution containing 2% Zoletil^®^ 50 (VirBac, Carros Cedes, France) and 1.5% xylazine (InterChemie, Venray, The Netherlands) at a dose of 1 mL/kg and then placed in a stereotaxic apparatus. The skin and aponeurosis on the head were dissected, and the skull was perforated 2.0 mm posterior to the bregma and 3.5 mm lateral to the sigma [65]. The brain tissue in the area of skull perforation was destroyed and superficial cerebral veins on the corresponding side were damaged. To standardize the volume of spilled blood, 20 μL of autologous blood taken from the sublingual vein was slowly injected into the destructed intern capsule. The wound was sprinkled with Ceftriaxone powder (Biosintez PJSC, Penza, Russia) and then sutured. One intravenous injection of 100 μL of neuroprotective composition was carried out in the tail vein of rats 1 h after ICH modeling. Rats that died within the first 24 h after ICH modeling were excluded from the study. There were no such rats in this study.

Twenty-two rats were used for comparison of neuroprotective activity of the secretomes of pMSCs and iMSCs: 6 rats—negative control, administration of empty DMEM-LG cell culture medium, 6 rats—10-fold concentrated pMSC secretome (pMSC10x), 10 rats—10-fold concentrated iMSC secretome (iMSC10x).

To evaluate the neuroprotective activity of the secretomes of genetically modified iMSCs, an additional 40 rats were used, with 5 rats in each of the 8 groups. The groups included a negative control (DMEM-LG), a positive control (non-concentrated iMSC secretome, iMSC1x), and the experimental groups: non-concentrated secretomes from genetically modified iMSCs (1x) overexpressing BDNF and uPA, constitutively (cBU) or inducibly (iBU); overexpressing anti-VHL shRNA, constitutively (cSh) or inducibly (iSh); or their combinations (via mixing of ready secretomes at the post-production stage) (cBU + cSh or iBU + iSh).

The sample size of the experimental groups was established experimentally on the basis of our earlier experiments [4,9], according to which 5 animals per experimental group was the minimum group sample size that allowed one to evaluate the neuroprotective activity of a substance in the model of intracerebral hemorrhage using such a sensitive method as MRI. To reduce animal suffering, we used the minimum possible number of animals.

The rats were numbered and assigned to experimental groups randomly using a random number generator. The experimenter (Stalik S. Dzhauari) who operated on the animals and administered the substances (solutions) and conducted the survival assessment and neurological testing was blinded to which group the animal belonged to and which substance he administered. The morphology specialist and the MRI operator were blinded to which group the animal belonged. Modeling of intracerebral hemorrhage in rats from the control and treatment groups was alternated and performed on the same day (all rats were operated on during several days). Rats from different groups lived in the same cage (3 rats per cage).

### 4.9. Assessment of Animal Survival and Neurological Status

The animals were observed for 14 days after ICH modeling, with daily recording of the percentage of survived rats and assessment of their neurological status at the 3rd and 10th days after ICH modeling using the Stroke index McGraw scale modified for rodents by I.V. Gannushkina [66,67]. Briefly, “visually healthy animals” had no signs of neurological deficit. Animals with signs of torpor, limb weakness, tremor, ptosis, and/or semi-optosis were classified as “slightly affected” animals. Animals with limb paresis and/or paralysis, impaired coordination, or in coma were classified as “severe affected” animals.

### 4.10. MRI

MR images were obtained 14 days after ICH modeling using the Clinscan 7T system (Bruker Biospin, Billerica, MA, USA) equipped with a coil for rat brain analysis with TurboS Spin Echo and signal suppression from adipose tissue. Coronal projections were obtained using the following parameters: TR (repetition time) = 5220 ms; TE (time to echo) = 53 ms; signal echo duration = 9; baseline resolution 230 × 320; FoV (field of view) = 32 × 40 mm; slice thickness = 0.5 mm; slice spacing = 0.75 mm. Transverse projections were obtained using the following parameters: TR = 4000 ms; TE = 40 ms; signal echo duration = 9; baseline resolution 288 × 320; FoV = 40 × 40 mm; slice thickness = 0.5 mm; and slice spacing = 0.6 mm. The brain lesion volume for each animal was calculated using the following formula: Lesion volume (mm^3^) = Sum [Lesion Area (mm^2^)] * (Slice Thickness (mm) + Slice Spacing (mm)).

### 4.11. Histologic Examination of Brain Slices

For histological confirmation of brain tissue damage, rats were sacrificed 14 days after ICH modeling by inhalation of a CO_2_-air gas mixture. The brain was extracted, fixed in 4% formaldehyde solution, and embedded in paraffin. Brain slices crossing the lesion focus were deparaffinized and stained with hematoxylin-eosin.

### 4.12. Statistics

Statistical analysis was performed using SigmaPlot11.0 software (Systat Software, Inc., Erkrath, Germany). Numerical data were assessed for normality of distribution using the Kolmogorov–Smirnov criterion. Differences between groups were analyzed using Student’s *t*-test (for pairwise comparisons) or analysis of variance (ANOVA): Newman–Keuls and Sidak–Holm tests for multiple comparisons if the distribution was normal. Analysis of variance (ANOVA) by rank (Dunn’s criterion) was used to compare groups of data with a non-normal distribution. Data are presented as the mean ± standard deviation or median (25%; 75%) depending on the distribution. Differences between groups were considered significant at *p* < 0.05 for all types of statistical analysis performed.

## 5. Conclusions

MSC secretome contains molecules with both synergistic and antagonistic activities and the resulting therapeutic effect of the secretome is largely determined by the ratio and interactions of these molecules with each other and with the recipient’s cells and molecules. The MSC secretome is a promising platform for developing drugs for regenerative biomedicine, and stabilization of its qualitative and quantitative composition can be achieved through the immortaliza-tion or genetic modification of producer cells. The secretomes of immortalized MSC cultures represents a promising and commercially viable alternative to that of primary isolated MSCs. The therapeutic properties of the secretome of genetically modified MSC cultures can vary significantly—from promoting regeneration to inducing toxic effects—depending on the specific modifications made to the producer cells. This variability necessitates careful and thorough evaluation of the secretome’s properties in each case.

## Figures and Tables

**Figure 1 ijms-26-06697-f001:**
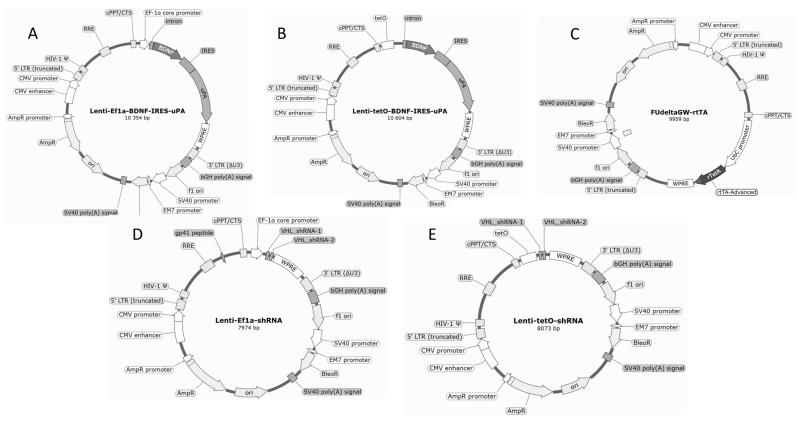
Maps of genetic lentiviral constructs used to increase the expression levels of BDNF, uPA, and pro-angiogenic factors in iMSC cultures. (**A**,**B**) Vectors for constitutive and inducible expression of BDNF and uPA protein combination, respectively; (**C**) Vector encoding a sensor for tetracycline/doxycycline [29]; (**D**,**E**) Vectors for constitutive and inducible expression of shRNA to VHL mRNA, respectively.

**Figure 2 ijms-26-06697-f002:**
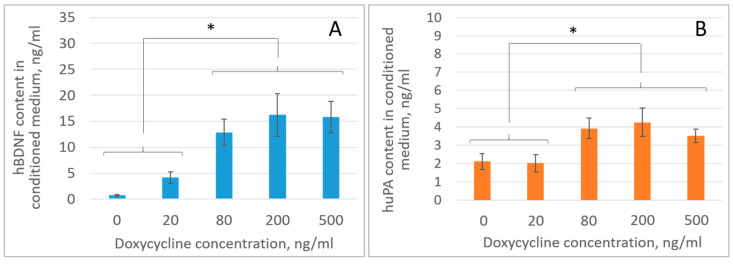
Determination of optimal doxycycline concentration for inducible expression of human (**A**) BDNF and (**B**) uPA in the cultures of iMSCs genetically modified with Lenti-tetO-BDNF-IRES-uPA and FuDeltaGW-rtTA vectors [29]. * *p* < 0.05, *n* = 3 (ANOVA, Newman–Keuls test).

**Figure 3 ijms-26-06697-f003:**
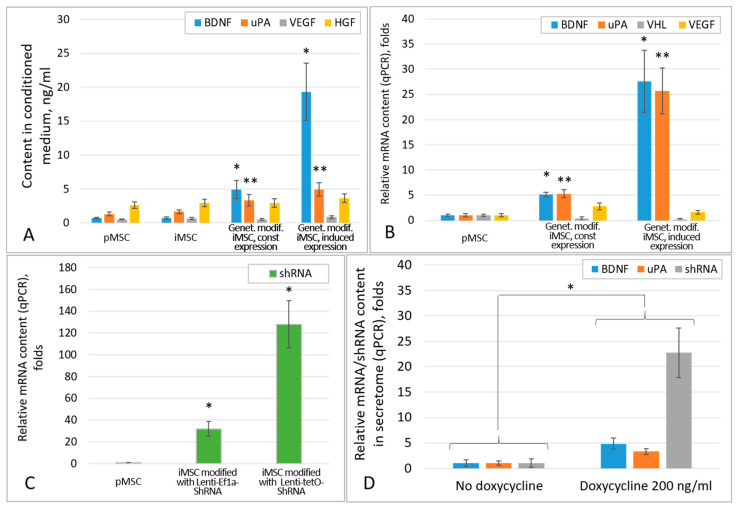
Comparative analysis of the efficiency of BDNF, uPA, VEGF, HGF, and anti-VHL shRNA expression by pMSC, iMSC, and genetically modified iMSC cultures. (**A**) The contents of BDNF and uPA or VEGF and HGF proteins in the culture medium of iMSCs genetically modified by BDNF-uPA or shRNA expression cassettes, respectively, according to ELISA data; (**B**) The mRNA contents of BDNF and uPA or VHL and VEGF in the lysates of iMSCs genetically modified by BDNF-uPA and shRNA expression cassettes, respectively, according to qPCR data; (**C**) shRNA to VHL mRNA contents in iMSC lysates by qPCR; (**D**) Relative levels of BDNF and uPA or anti-VHL shRNA representation in the secretome of iMSCs genetically modified with BDNF-uPA or shRNA inducible expression cassettes, respectively, after induction with 200 ng/mL doxycycline. (**A**–**C**): *, ** *p* < 0.05 vs. corresponding values in the pMSC group, *n* = 3 (ANOVA, Newman–Keuls test). (**D**): * *p* < 0.05 (*n* = 3, Student’s *t*-test).

**Figure 4 ijms-26-06697-f004:**
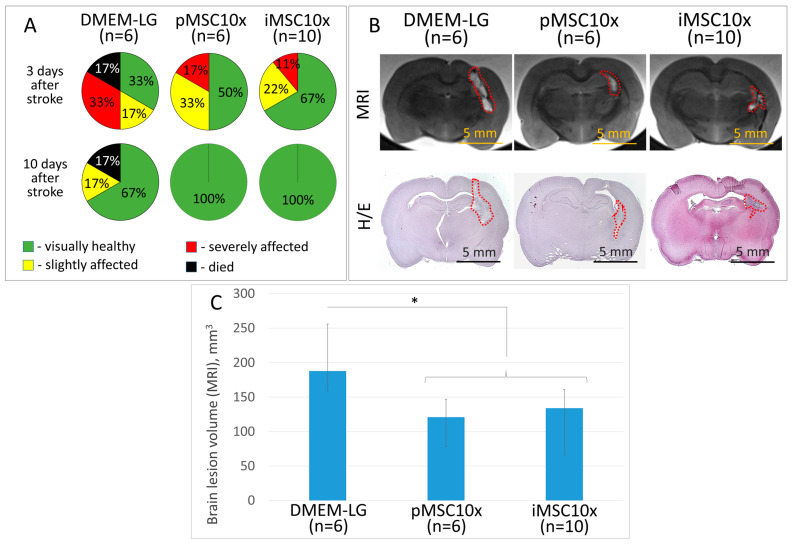
Results of evaluation of neuroprotective activity of iMSC secretome in the ICH model. (**A**) Severity of neurological deficits in experimental animals 3 and 10 days after ICH modeling (*p* ≥ 0.11, n ≥ 6, 2 × 2 Chi square test); (**B**) Samples of brain lesion foci 14 days after ICH (MRI, histochemical staining), H/E—hematoxylin/eosin; (**C**) Quantitative assessment of brain lesion volume according to MRI. Data are presented as median (25%; 75%). DMEM-LG—empty culture medium. pMSC10x and iMSC10x—10-fold concentrated secretome of primary isolated or immortalized human MSCs, respectively. (* *p* < 0.05, n ≥ 6 (ANOVA by rank, Dunn’s criterion).

**Figure 5 ijms-26-06697-f005:**
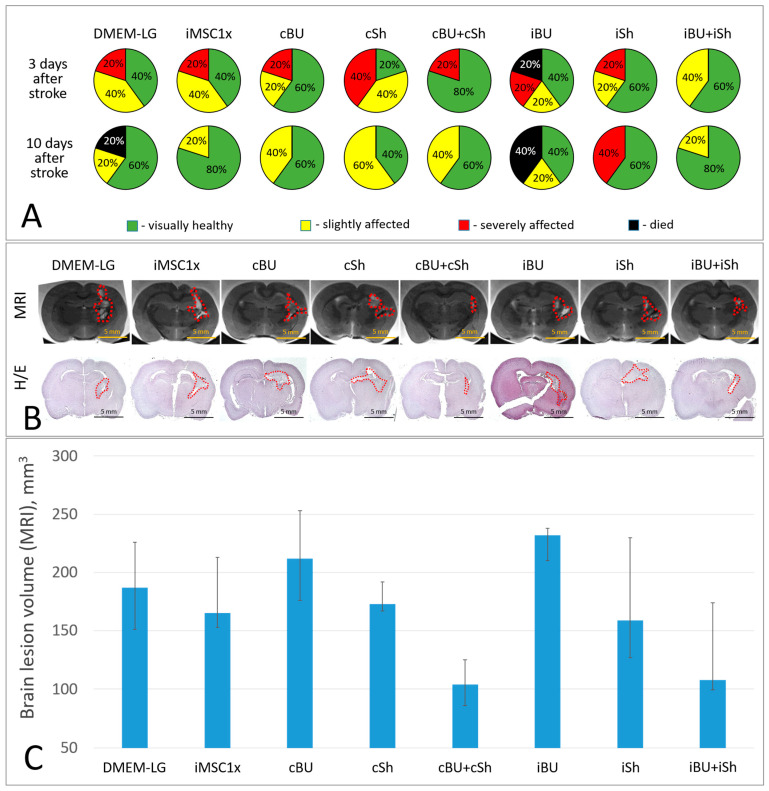
Results of evaluation of neuroprotective activity of genetically modified iMSC secretome in the ICH model. (**A**) Severity of neurological deficits in experimental animals 3 and 10 days after ICH modeling (*p* ≥ 0.44, *n* = 5, 2 * 2 Chi square test); (**B**) Samples of brain lesion foci 14 days after ICH modeling (MRI, histochemical staining), H/E—hematoxylin/eosin; (**C**) Quantitative assessment of brain lesion volume according to MRI. Data are presented as median (25%; 75%). *n* = 5, ANOVA by rank, Dunn’s criterion. iMSC1x—non-concentrated secretome of immortalized MSCs, iBU—the secretome of MSCs inducibly expressing BDNF and uPA, iSh—the secretome of MSCs inducibly expressing shRNAs to VHL mRNA, cBU—the secretome of MSCs constitutively expressing BDNF and uPA, cSh—the secretome of MSCs constitutively expressing shRNAs to VHL mRNA.

**Table 1 ijms-26-06697-t001:** BDNF, uPA, VEGF, and HGF contents in the unconcentrated (1x) secretomes of pMSCs, iMSCs, and genetically modified iMSCs (*n* = 3).

	pMSC	iMSC	cBU	cSh	cBU + cSh	iBU	iSh	iBU + iSh
BDNF	0.70 ± 0.16	0.73 ± 0.16	4.90 ± 1.31	0.81 ± 0.18	3.25 ± 0.45	19.32 ± 4.2	0.68 ± 0.15	9.43 ± 1.41
uPA	1.30 ± 0.28	1.65 ± 0.29	3.33 ± 0.84	1.39 ± 0.33	2.28 ± 0.40	4.94 ± 0.99	1.61 ± 0.34	2.97 ± 0.45
VEGF	0.53 ± 0.06	0.66 ± 0.19	0.71 ± 0.17	0.53 ± 0.11	0.59 ± 0.15	0.65 ± 0.19	0.86 ± 0.18	0.76 ± 0.11
HGF	2.62 ± 0.49	2.93 ± 0.57	2.38 ± 0.56	2.96 ± 0.63	2.61 ± 0.31	2.67 ± 0.59	3.61 ± 0.63	3.07 ± 0.38

## Data Availability

Data are available upon request.

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
