# Peer review of "Overexpression of BDNF and uPA Combined with the Suppression of Von Hippel–Lindau Tumor Suppressor Enhances the Neuroprotective Activity of the Secretome of Human Mesenchymal Stromal Cells in the Model of Intracerebral Hemorrhage"

_ijms, 2025, doi:10.3390/ijms26146697_

Round 1
Reviewer 1 Report
Comments and Suggestions for Authors
This manuscript titled "Overexpression of BDNF and uPA combined with the suppression of Von Hippel–Lindau tumor suppressor enhances the neuroprotective activity of the secretome of human mesenchymal stromal cells in the model of intracerebral hemorrhage" presents an innovative and well-structured study exploring genetic enhancement strategies to improve the therapeutic efficacy of MSC-derived secretomes in a preclinical model of intracerebral hemorrhage. The authors employ both overexpression of neurotrophic and angiogenic factors (BDNF, uPA) and suppression of VHL to modulate the secretome content, offering a potentially scalable and clinically translatable cell-free therapy. The experimental design is generally sound, and the data are well presented. However, several aspects related to mechanistic interpretation, statistical analysis, and methodological transparency require clarification and improvement before the manuscript can be considered for publication.
Title and Abstract
- Title (Lines 1–2): The title is comprehensive but too long. Consider shortening it for better impact, e.g., “Genetic Enhancement of MSC Secretome Improves Neuroprotection in Intracerebral Hemorrhage”.
- Lines 22–24: “...although it also requires 10-fold concentration” — This sentence could be clearer. Consider rephrasing to “...yet still requires 10-fold concentration to achieve neuroprotective efficacy.”
- Lines 30–33: You mention that individual modifications (BDNF, uPA, VHL suppression) had no or toxic effects. This is a critical point — consider briefly explaining why this may be the case in one sentence for context.
Introduction
- Lines 46–49: The background and rationale are well-explained. Good inclusion of previous work on secretome.
- Line 66: Citation [9] refers to the need for 5–10x concentration — a quantitative reference is helpful here but consider clarifying whether this is based on in vivo efficacy thresholds or in vitro potency.
- Lines 93–112: The dual approach (forced gene expression vs. VHL suppression) is innovative and well-described. However, the rationale for selecting VHL suppression as a proxy for HIF-1a activation would benefit from additional mechanistic context or referencing relevant HIF-1a targets beyond VEGF (e.g., EPO, angiopoietin).
Results
- Lines 125–127: The baseline secretome concentrations are reported, but no standard deviation is given for BDNF in this line, unlike others. Please ensure all values are complete and consistent.
- Lines 136–144 (Fig. 2): The plateau effect with doxycycline is well-demonstrated. You may consider noting whether cell viability was assessed at high doxycycline doses (e.g., 500 ng/mL).
- Lines 153–162: Impressive increase in BDNF upon induction (~22-fold). For uPA, the 2-fold increase is more modest — consider commenting on why the relative upregulation differs.
- Lines 171–174: The data on slowed proliferation due to constitutive expression is important. Please clarify if this was quantified by population doubling time or viability assays (e.g., MTT).
- Lines 183–193 (Fig. 4): The comparison between pMSC10x and iMSC10x is strong. However, the survival and MRI data should be supported by statistical power analysis or a statement about n size adequacy.
- Line 194: State whether lesion volumes are based on 3D reconstructions or 2D slice summations.
- Lines 219–223: You describe combined use of cBU+cSh as most effective. However, no dose-response or isobologram analysis is provided. Consider including or suggesting future studies to determine if synergy exists.
- Lines 229–239: The toxic effect of high-dose BDNF/uPA is critical. It would be helpful to include histological markers of apoptosis or inflammation to support the mechanistic interpretation.
- Lines 248–251: Clarify whether secretome combinations (iBU+iSh) were simply mixed post-production or if co-culture was used. This affects translational relevance.
Discussion
- Lines 285–292: Solid rationale for using immortalized MSCs. However, add a note on regulatory considerations for using genetically modified vs. immortalized lines in clinical applications.
- Lines 328–344: The possibility of toxic effects from pro-BDNF is intriguing. You should cite studies showing p75NTR pathway activation in similar contexts (e.g., [13,14] may not be sufficient).
- Lines 367–379: The concept of synergy between neurotrophic and angiogenic factors is compelling but speculative. Add a sentence suggesting next steps, such as proteomic analysis or single-cell transcriptomics, to dissect this interaction.
Materials and Methods
- Lines 390–402: Good details on cell lines and markers. However, include the passage numbers used for each iMSC group at time of secretome harvest for reproducibility.
- Line 420: For viral transduction, report transduction efficiency or any selection strategy (e.g., antibiotic resistance or FACS).
- Line 474: Secretome concentration by ultrafiltration is mentioned. Was protein loss assessed during this step? If not, recommend noting this limitation.
- Lines 509–516: ICH model description is detailed. Please report if the same surgeon performed all procedures and if animals were randomized/blinded.
- Line 550: Consider specifying which scoring scale was used to define “severe” vs. “mild” neurological deficit — currently very general.
Comments on the Quality of English Language
The English could be improved to more clearly express the research.
Author Response
Dear Reviewer, Thank you very much for your time and valuable comments. Here below, we provide a point-by-point response to your comments. All changes to the original manuscript are highlighted in green.
Title and Abstract
* Title (Lines 1–2): The title is comprehensive but too long. Consider shortening it for better impact, e.g., “Genetic Enhancement of MSC Secretome Improves Neuroprotection in Intracerebral Hemorrhage”.
- Thank you for the suggested title. We also considered this option. However, it is too general and does not fully reflect the results obtained. In our study, we came to the conclusion that only certain types of genetic modifications increase neuroprotective activity. An increase in the concentration of a specific neurotrophic or proangiogenic factor does not always stimulate the regenerative process, as one might assume mechanistically. This is probably due to non-optimal concentration or balance of growth factors, the absence of auxiliary or the presence of inhibitory molecules, the absence of the necessary receptors on the target cells or the suppression of signaling cascades. All of this requires further study. One of the key findings of our study is "The therapeutic properties of the secretome of genetically modified MSC cultures can vary significantly—from promoting regeneration to inducing toxic effects—depending on the specific modifications made to the producer cells. This variability necessitates careful and thorough evaluation of the secretome's properties in each case." In our opinion the original title more accurately reflects the results obtained and we would like to leave it unchanged.
* Lines 22–24: “...although it also requires 10-fold concentration” — This sentence could be clearer. Consider rephrasing to “...yet still requires 10-fold concentration to achieve neuroprotective efficacy.”
- Thank You for the consideration! We have made the correction to the manuscript.
* Lines 30–33: You mention that individual modifications (BDNF, uPA, VHL suppression) had no or toxic effects. This is a critical point — consider briefly explaining why this may be the case in one sentence for context.
- Presumably, this may be due to an imbalance in the representation of individual growth factors in the secretome of genetically modified MSCs, which individualy may lead to undesirable effects in damaged nervous tissue, such as increased permeability of the blood-brain barrier (under the influence of proangiogenic factors) or neural cells apoptosis (due to an excess of neurotrophic factors). We have added this information to the Abstract section.
Introduction
* Lines 46–49: The background and rationale are well-explained. Good inclusion of previous work on secretome.
- Thank you! We have tried to briefly bring up the existing problem based on the data of our previous research and reflect the course of our thoughts.
* Line 66: Citation [9] refers to the need for 5–10x concentration — a quantitative reference is helpful here but consider clarifying whether this is based on in vivo efficacy thresholds or in vitro potency.
- Concentration of the secretome by 5-10-fold is required to achieve noticable neuroprotective activity in the intracerebral hemorrhage model in rats in vivo, which is described in citation [9]. Similar results were obtained by us and our colleagues in other models: treatment of male infertility caused by cryptorchidism in vivo [10.1016/j.biopha.2018.10.174], a model of direct neuroprotection in vitro [10.3390/pharmaceutics13122031], etc. It can be said that in most cases, concentration of the secretome is necessary to achieve the desired therapeutic effect. This information was added to the manuscript.
* Lines 93–112: The dual approach (forced gene expression vs. VHL suppression) is innovative and well-described. However, the rationale for selecting VHL suppression as a proxy for HIF-1a activation would benefit from additional mechanistic context or referencing relevant HIF-1a targets beyond VEGF (e.g., EPO, angiopoietin).
- This information was added to the manuscript.
Results
* Lines 125–127: The baseline secretome concentrations are reported, but no standard deviation is given for BDNF in this line, unlike others. Please ensure all values are complete and consistent.
- Concentrations of BDNF and other growth factors are indicated in the article: in the text and in Table 1. Thus, the concentration of BDNF in the secretome of unmodified primary MSCs is 0.7 ± 0.16 ng/mL, and in the secretome of immortalized MSCs - 0.73 ± 0.16 ng/mL.
* Lines 136–144 (Fig. 2): The plateau effect with doxycycline is well-demonstrated. You may consider noting whether cell viability was assessed at high doxycycline doses (e.g., 500 ng/mL).
- Yes, you are right. In this study, we did not attach importance to this and did not investigate this aspect, because we did not observe visible cell death in such MSC cultures. MSCs are considered to be a fairly stable cell culture to various toxic and biological effects. According to the literature, doxycycline concentrations up to 2 μg/ml do not have a toxic effect and do not affect the differentiation potential of human MSCs [10.1016/j.joca.2012.11.010]. This notion was added to the manuscript.
* Lines 153–162: Impressive increase in BDNF upon induction (~22-fold). For uPA, the 2-fold increase is more modest — consider commenting on why the relative upregulation differs.
- This can be explained by a combination of two facts. The first fact is a high base level of urokinase expression (~ 3 folds higher than BDNF), which masks minor changes. The second is the structure of the expression cassette of the Lenti-tetO-BDNF-IRES-uPA vector, where BDNF and uPA are encoded within a single bicistronic RNA. In such a construct BDNF translation from the bicistronic RNA occurs via a 5'-cap-dependent mechanism, and uPA translation via an IRES-dependent mechanism. The 5'-cap-dependent translation mechanism is 10-30 times more productive than the IRES-dependent mechanism [10.1016/j.virusres.2008.06.004; 10.1124/jpet.119.261594]. We have added this information to the manuscript.
* Lines 171–174: The data on slowed proliferation due to constitutive expression is important. Please clarify if this was quantified by population doubling time or viability assays (e.g., MTT).
- We agree with you, but we did not focus on this aspect. This information was obtained as a result of visual observation of cell cultures during their passaging. Genetically modified cell cultures reached a monolayer 2-3 times slower than unmodified immortalized MSCs (or early passage pMSCs), which corresponds to an increase in the population doubling time rather than their reduced survival or excessive death. Within this study, this is only an observation that attracted our attention. This fact requires more careful and thorough study. We assume that changes in the proliferation rate of the producer culture are largely determined by the nature of the cell culture itself and the kind of genetic modification.
* Lines 183–193 (Fig. 4): The comparison between pMSC10x and iMSC10x is strong. However, the survival and MRI data should be supported by statistical power analysis or a statement about n size adequacy.
- This information is reflected in the Materials and Methods section: 4.8. Intracerebral hemorrhage model: "22 rats were used for comparison of neuroprotective activity of the secretomes of pMSC and iMSC: 6 rats - negative control, administration of empty DMEM-LG cell culture medium, 6 rats - 10-fold concentrated pMSC secretome (pMSC10x), 10 rats - 10-fold concentrated iMSC secretome (iMSC10x)."
We have added the number of animals for each group to Figure 4. The sample size is not sufficient to detect statistically significant differences between the groups in neurological status. There is a trend towards differences between the control group and the MSC groups (pMSC10x and iMSC10x).
* Line 194: State whether lesion volumes are based on 3D reconstructions or 2D slice summations.
- The brain lesion volume for each animal was calculated using the following formula: Lesion volume (mm3) = Sum (Lesion Area (mm2)) * (Slice Thickness (mm) + Slice Spacing (mm)).
This information was added to the Materials and Methods section: 4.10. MRI
* Lines 219–223: You describe combined use of cBU+cSh as most effective. However, no dose-response or isobologram analysis is provided. Consider including or suggesting future studies to determine if synergy exists.
- Yes, we agree with you. The therapeutic activity of the secretome mixtures (iBU+iSh and cBU+cSh) requires further studies of their effecteviness in dose-dependence and isobologram analysis, which may be the prospective of this research. We have added this information to the Discussion section.
* Lines 229–239: The toxic effect of high-dose BDNF/uPA is critical. It would be helpful to include histological markers of apoptosis or inflammation to support the mechanistic interpretation.
- Yes, we agree with you. Hematoxylin/eosin staining of rat brain sections revealed an increased volume of necrotic tissue in the cBU and iBU groups, but no evidence of significant leukocyte infiltration. Establishing the mechanism of the toxic effect of high-dose BDNF/uPA requires additional studies. Possible mechanisms include increased neuroinflammation, impaired blood-brain barrier function, or direct toxic effects of excessive amounts of growth factors. Our previously published data reveal that uPA can stimulate microglial activation when administered intracerebrally [10.3390/biomedicines10061346]. Which effect (pro-regenerative or toxic) will be predominant is apparently determined by the concentration of uPA and the duration of its action. We have added this information to the Discussion section.
* Lines 248–251: Clarify whether secretome combinations (iBU+iSh) were simply mixed post-production or if co-culture was used. This affects translational relevance.
- Secretome combinations iBU+iSh and cBU+cSh were obtained via mixing of ready secretomes iBU and iSh or cBU and cSh, respectively, at the post-production stage. This information was added to materials and methods section.
Discussion
* Lines 285–292: Solid rationale for using immortalized MSCs. However, add a note on regulatory considerations for using genetically modified vs. immortalized lines in clinical applications.
- Yes, you are right that immortalized and genetically modified MSCs attract special attention of regulatory authorities due to their possibility of tumor transformation after transplantation. Any such line requires careful and long-term study and its application must be limited to strict indications. At the same time, the use of the secretome of immortalized or genetically modified lines is devoid of such disadvantages, since it does not contain cells capable of transformation. As any biotechnological product or drug, the secretome must be just effective and safe. In this paragraph, I am not talking about immortalized MSCs themselves, but about iMSC-derived products (secretome). I am afraid that including a sentence about ​​regulatory restrictions on immortalized and genetically modified MSCs (regulation is not the main aspect of this work) may disrupt the logic of the presentation and complicate perception of the idea of the study. We talk about the need to carefully study the properties of the secretome of genetically modified cells in the Discussion section: "This variability necessitates careful and thorough evaluation of the secretome's properties in each case."
* Lines 328–344: The possibility of toxic effects from pro-BDNF is intriguing. You should cite studies showing p75NTR pathway activation in similar contexts (e.g., [13,14] may not be sufficient).
- Mature BDNF and its proform are secreted together, and increased expression of the BDNF gene (encoding the pro-BDNF) may alter the natural ratio of BDNF to pro-BDNF due to the inability of cellular proteases (Furin, PC5-B, PC7, PC4, PACE4, PC5, and SKI-1) to activate excessive pro-BDNF before secretion. The fact that pro-BDNF reveales prominent toxic activity is confirmed by many studies [10.1016/j.brainresbull.2025.111338; 10.1038/s41398-021-01667-2; 10.1177/1759091420930865]. These references were added to the manuscript.
* Lines 367–379: The concept of synergy between neurotrophic and angiogenic factors is compelling but speculative. Add a sentence suggesting next steps, such as proteomic analysis or single-cell transcriptomics, to dissect this interaction.
- Thanks for the recommendation! We have added this sentence: "The investigation of specific mechanisms of possible cooperative activity of neurotrophic and angiogenic factors requires additional studies involving inhibitory, transcriptomic or signalomic analysis." This assumption is supported by previously published works: 10.14336/AD.2021.1121 (VEGF increases blood brain barrier); 10.1111/cns.13082 and 10.1016/j.bbadis.2017.01.023 (plasmin/plasminogen activators stimulate maturation of neurotrophic factors), etc.
Materials and Methods
* Lines 390–402: Good details on cell lines and markers. However, include the passage numbers used for each iMSC group at time of secretome harvest for reproducibility.
- We absolutely agree with you that the passage of a cell culture can determine the properties of its secretome. This information is provided in the Materials and Methods section: 4.5. Obtaining MSC secretome: "...subconfluent cultures of human pMSC (5-7 passages), iMSC (13-15 passages) or genetically modified iMSC (12-14 passages) were thoroughly washed with Hanks' solution (PanEco, #P020P) and then cultured for 7 days in DMEM low glucose medium ...". We perform conditioning of pMSC cultures at passages 5-7, since they still contain relatively few senescent cells, and their secretomes have a low content of SASP components. Genetic modification, cell selection and obtaining a large population of immortalized/genetically modified cells require a lot of time and several passages, so the secretome of iMSC or genetically modified iMSC cultures can be obtained several passages later than for pMSC (12-15 passages). The used passages within each of these groups allow obtaining a secretome of comparable quality. We have conducted a thorough comparative study of the properties of the pMSC and iMSC secretomes in dynamics (data not published, under consideration for publication in the European Journal of Pharmaceutics and Biopharmaceutics), which showed that the iMSC secretome retains its qualitative and quantitative composition and biological activity for at least 20 passages, when the content of SASP components begins to increase. A similar increase in the content of SASP components for pMSC cultures begins at passages 8-9. The results can be provided upon request.
* Line 420: For viral transduction, report transduction efficiency or any selection strategy (e.g., antibiotic resistance or FACS).
- Immortalized MSC cultures (iMSC) were previously obtained and characterized by our team [10.3390/ijms25042421], selection of the transduced pMSC cultures during the immortalization procedure was performed using puromycin.
The obtained iMSC line was used for additional genetic modification with the overexpression cassettes encoding genes of growth factors or shRNAs to the VHL mRNA. Within this experiment we didn't perform any additional round of selection, since the vectors did not contain additional selection cassettes in order to increase the efficiency of LVP assembly and transduction. After transduction, the total genetically modified iMSC population was used.
For optimal lentiviral transduction, we followed the previously developed protocol for lentiviral particle assembly and transduction: we used the recommended iMSC confluent and LVP titer, all LVP assembly procedures were performed in parallel, iMSC transduction with different LVPs was performed 4 times and in parallel. The lentiviral particles used contained a short genome (<6 kbp), which ensures (in combination with the above conditions) a high level of transduction [10.1089/104303401753153947]. As used lentiviral vectors didn't contain any selection marker transduction efficiency was controlled by co-transduction with LVPs encoding the BFP (blue fluorescent protein) gene in a ratio LVP(BU/shRNA) : LVP(BFP) = 4 : 1. In a week since the 4 rounds of transduction, fluorescence in the BFP channel in all genetically modified iMSC cultures exceeded 80%. This information was added to the Materials and Methods section 4.4. Genetic modification of MSC.
* Line 474: Secretome concentration by ultrafiltration is mentioned. Was protein loss assessed during this step? If not, recommend noting this limitation.
- No, protein loss in the filtrate was not assessed. This information has been added to the corresponding section of "Materials and Methods" as one of the limitations of the study. We used secretome concentration only when comparing the neuroprotective activity of pMSC and iMSC secretomes. This procedure was performed in parallel using the concentrating cartridges from the same manufacturer. Concentration of the secretomes of genetically modified iMSC cultures was not required, since they contained increased concentrations of growth factors.
* Lines 509–516: ICH model description is detailed. Please report if the same surgeon performed all procedures and if animals were randomized/blinded.
- "The same surgeon (Stalik S. Dzhauari) modeled ICH in all groups. He operated the animals, administered the substances (solutions), conducted the survival assessment and neurological testing. He was blinded to which group the animal belonged to and which substance he administered. The morphology specialist and the MRI operator were blinded to which group the animal belonged to." This information was added to the Materials and Methods section 4.8. A model of intracerebral hemorrhage.
* Line 550: Consider specifying which scoring scale was used to define “severe” vs. “mild” neurological deficit — currently very general.
- To assess the neurological status of animals, we used the Stroke-index McGraw scale modified for rodents by I.V. Gannushkina. Briefly, "visually healthy animals" had no signs of neurological deficit. Animals with signs of torpor, limb weakness, tremor, ptosis and/or semi-optosis were classified as "slightly affected" animals. Animals with limb paresis and/or paralysis, impaired coordination, or in coma were classified as "severely affected" animals. This scale also has a digital representation, but due to the relatively small sample, it can give significant scatter and is not very informative. The proposed representation method of neurological deficits is more visual for characterizing the neurological status of experimental animals. In Fig 4A and Fig 5A, digitalized data are already presented - the diagrams reflect the proportion of animals (healthy, dead, with mild or severe neurological impairment) in each group for two time points. Comparison of the obtained data was carried out using the 2 * 2 chi square test. These data were reflected in the figure captions. Statistically significant differences were not found between these groups due to the small samples (although tendencies towards such differences were observed) - this is one of the limitations of using the 2*2 chi square test.

Reviewer 2 Report
Comments and Suggestions for Authors
The article entitled “Overexpression of BDNF and uPA combined with the suppression of Von Hippel–Lindau tumor suppressor enhances the neuroprotective activity of the secretome of human mesenchymal stromal cells in the model of intracerebral hemorrhage” showed the the secretome of MSC cultures with combined hyperexpression of BDNF and uPA and suppressed expression of Von Hippel-Lindau tumor suppressor even without additional concentration reduced the severity of neurological disorders and decreased brain lesion volume in the ICH model.The data show that genetic modification of MSC cultures can be promising biopharmaceutical substances. There still some possible points for improvement or further consideration.
- In Fig 5C, are there some Significant differences between different group? It should be labeled.
- In Fig 4A and Fig 5A, the everity of neurological deficits should be also digitization and perform the statistic analysis between different group.
- What do iMSC,pMSC, iBU,iSh,cBU,cSh mean? They should also be make clear in the figure legends.
Author Response
Dear Reviewer, Thank you very much for your time and valuable comments. Here below, we provide a point-by-point response to your comments. All changes to the original manuscript are highlighted in green.
* In Fig 5C, are there some Significant differences between different group? It should be labeled.
- No significant differences in the volume of brain lesion foci between studied groups was observed due to the small sample size and multiple experimental groups; however, there was a trend toward lower brain lesion volume in the cBU+cSh group. That was noted in the text.
Pairwise comparisons of the cBU+cSh and iBU+iSh groups to the control group (DMEM-LG) reveal statistically significant differences that are masked in multiple groups comparison.
* In Fig 4A and Fig 5A, the everity of neurological deficits should be also digitization and perform the statistic analysis between different group.
- In Fig 4A and Fig 5A, digitalized data are already presented - the diagrams reflect the proportion of animals (healthy, dead, with mild or severe neurological impairment) in each group for two time points. Comparison of the obtained data was carried out using the 2 * 2 chi square test. These data were reflected in the figure captions. Statistically significant differences were not found between these groups due to the small samples (although tendencies towards such differences were observed) - this is one of the limitations of using the 2*2 chi square test.
To assess the neurological status of animals, we used the Stroke-index McGraw scale modified for rodents by I.V. Gannushkina. Briefly, "visually healthy animals" had no signs of neurological deficit. Animals with signs of torpor, limb weakness, tremor, ptosis and/or semi-optosis were classified as "slightly affected" animals. Animals with limb paresis and/or paralysis, impaired coordination, or in coma were classified as "severely affected" animals. This scale also has a digital representation, but due to the relatively small sample, it can give significant scatter and is not very informative. The proposed representation method of neurological deficits is more visual for characterizing the neurological status of experimental animals.
* What do iMSC,pMSC, iBU,iSh,cBU,cSh mean? They should also be make clear in the figure legends.
- iMSC,pMSC,iBU,iSh,cBU,cSh were deciphered in the figure legends. iMSC - the secretome of immortalized MSCs, pMSC - the secretome of primary MSCs, iBU - the secretome of MSCs inducibly expressing BDNF and uPA,iSh - the secretome of MSCs inducibly expressing shRNAs to VHL mRNA,cBU - the secretome of MSCs constitutively expressing BDNF and uPA,cSh - the secretome of MSCs constitutively expressing shRNAs to VHL mRNA.

Round 2
Reviewer 1 Report
Comments and Suggestions for Authors
I am satisfied with the authors’ responses to my previous comments and appreciate the revisions they have made to address the concerns raised. The manuscript is now significantly improved, and I believe it meets the standards for publication. I have no further concerns, and I recommend the paper for acceptance in its current form.